# Constipation and Its Associated Factors among Patients with Dementia

**DOI:** 10.3390/ijerph17239006

**Published:** 2020-12-03

**Authors:** Chien-Liang Chen, Tzu-Ming Liang, Hsiu-Hui Chen, Yan-Yuh Lee, Yao-Chung Chuang, Nai-Ching Chen

**Affiliations:** 1Division of Nephrology, Kaohsiung Veterans General Hospital, Kaohsiung 813, Taiwan; cclchen@seed.net.tw; 2Department of Medicine, National Yang-Ming University School of Medicine, Taipei 112, Taiwan; 3Nutrition Therapy Department, Kaohsiung Chang Gung Memorial Hospital and Chang Gung University College of Medicine, Kaohsiung 813, Taiwan; dino09200@cgmh.org.tw; 4Physical Education, National Kaohsiung University of Science and Technology, Kaohsiung 807, Taiwan; evahhchen@gmail.com; 5Department of Physical Medicine and Rehabilitation, Kaohsiung Chang Gung Memorial Hospital, Kaohsiung 833, Taiwan; b9005009@cgmh.org.tw; 6Department of Neurology, Kaohsiung Chang Gung Memorial Hospital and Chang Gung University College of Medicine, Kaohsiung 833, Taiwan; ycchuang@adm.cgmh.org.tw; 7Department of Neurology, School of Medicine, College of Medicine, Kaohsiung Medical University, Kaohsiung 807, Taiwan

**Keywords:** dementia, Mini-Nutritional Assessment, constipation, fiber, water, diet diary record

## Abstract

Constipation is one of the most frequent non-motor problems in older adults. As constipation is commonly ignored by dementia patients, it is not usually reported on time. Constipation has a serious impact on the activity of daily living and quality of life in dementia patients. The relationships between constipation, demographic variables, and the nutritional status of patients with dementia remain unknown. This study aimed to assess the possible factors associated with constipation. This cross-sectional study was conducted at the Kaohsiung Chang Gung Memorial Hospital from January to November 2019. This hospital is a medical center and the main referral hospital of southern Taiwan, serving 3 million inhabitants. In total, 119 patients with dementia were evaluated using the Rome III diagnostic criteria for functional constipation. There were 30 patients with dementia included in the constipation group and 89 patients with dementia included in the no constipation group. Mini-Nutritional Assessment and 3-day diet diary records were employed. The clinical dementia rating score was used to evaluate the severity of dementia in patients of the outpatient clinic. Approximately 25.2% of dementia patients had constipation. Patients in the dementia with constipation group were older, had severer dementia, and displayed a lower water intake. After multivariable adjustment, low liquid consumption was the predictor of constipation among patients with dementia. The findings support the clinical recommendations to treat constipation with an increased liquid intake, but not exercise, in dementia patients.

## 1. Introduction

Dementia is a neurodegenerative disease with a progressive, irreversible, and long-term course. Its recent high prevalence rate in Western and Asian countries has a significant impact on modern society due to global aging [1]. The great majority of individuals with dementia are cared for by family members in the community. With the increasing number of individuals with dementia, several questions have been raised on how parents are being cared for. Currently, there is no cure for dementia; treatment is only focused on the care of dementia patients and allowing the person with dementia and their families to live well in the community. Caring for patients with dementia poses many challenges for families and caregivers. A meta-analysis study found that dementia family caregivers were significantly more stressed than non-dementia caregivers, and more likely to experience serious depressive symptoms and physical problems [2]. Overall, one-third of the caregivers (disregarding the care demand and care recipient in dementia stage) provided >14 h of care on a daily basis. Meanwhile, basic and instrumental activities of daily living (ADLs) impairments (e.g., bathing and toileting as daily basic life needs) (68%) were the most common concerns of most caregivers [3]. Understanding how to support the daily activities of dementia patients could be helpful to families, caregivers, and patients themselves.

Toileting, which includes bowel movements (stool passage) and urination, is a basic need, and constipation and urinary incontinence are common health problems among older adults, including those with dementia. Constipation is one of the most common health problems encountered in primary healthcare settings and occurring in the general population, as well as among older adults, especially those with dementia [3,4,5]. Constipation is characterized by the formation of hard stools, infrequent defecation (fewer than two times per week), and the prolonged and difficult evacuation of stools. Patients with persistent constipation have a lower quality of life and experience financial burdens owing to health costs [6,7,8,9]. Patients with dementia might not be able to express the pain and discomfort of the constipation orally and may display aggressive behavior instead, so the constipation will be untreated or the patient will be wrongly given an anti-psychotic drug [10,11,12]. This can make matters worse because the anti-psychotic drugs may cause constipation. Therefore, how to prevent constipation is very important. An ounce of prevention is worth a pound of cure.

The pathogenesis of constipation is multifactorial, with a focus on dietary behavior, such as low fiber consumption, a lack of adequate fluid intake, a lack of mobility, disturbance in the hormone balance, the side effects of medications with anti-cholinergic actions, and hemorrhoids [8,9,10,11,12]. However, constipation has never been isolated in patients with dementia. Given that epidemiological and clinical studies have reported an association of constipation in patients with dementia and that little is known about the risk factors of constipation, such as demographic factors and diet behaviors, this study aimed to assess the epidemiological data and possible factors associated with constipation.

## 2. Materials and Methods

### 2.1. Ethical Statement

A total of 119 patients with dementia in the Dementia Outpatient Clinic of Kaohsiung Chang Gung Memorial Hospital were enrolled in the study. Patients or primary caregivers who could not record all food consumed within a 72 h period were excluded from this study. All procedures performed in the study were in accordance with the 1964 Helsinki Declaration and its later amendments or comparable ethical standards. This study was approved by the Chang Gung Medical Foundation Institutional Review Board, and informed consent was obtained from all participants (CGMH-IRB No 201600385B0).

In the International Classification of Diseases, Tenth Edition (ICD-10) [13], dementia is defined as a disorder with deterioration in both memory and thinking, which is sufficient to impair the personal activities of daily living. The definition requires that the patient has deficits in thinking and reasoning, in addition to memory disturbance. The Diagnostic and Statistical Manual of Mental Disorders, Fourth Edition (DSM-IV) [14], defines dementia as a syndrome characterized by the development of multiple cognitive deficits, including memory impairment and at least one of the following cognitive disturbances: aphasia; apraxia; agnosia; and a disturbance in executive functioning.

The criteria included in the study were that the patients, supported by their caregivers, could receive all examinations and complete the records. The main caregiver needed to stay with the patient every day and have no subjective cognitive decline/complaint. The exclusion criteria for all subjects in the study were patients who had colorectal cancer or colorectal inflammation disease. The other exclusion criteria for enrolled subjects included end stage renal disease, tumor under hospice care, chronic pain with regular painkillers in use (morphine, ultracet, and nonsteroidal anti-inflammatory drugs), and hematologic, endocrine, and autoimmune diseases. 

### 2.2. Demographic and Neurobehavioral Assessments

All patients underwent general physical examinations and clinical interviews. Patients’ clinical data were obtained, including their age, sex, and comorbidities, including diabetes mellitus, hypertension, hyperlipidemia, coronary artery disease, and chronic kidney disease. 

A trained neuropsychologist administered the neurobehavioral tests, including the Mini-Mental State Examination (MMSE) [15] to assess their general intellectual function, and clinical dementia rating (CDR) scores [16] to evaluate the severity of patients’ cognitive impairment in the dementia integrated outpatient clinic.

### 2.3. Blood Collection and Analysis of the Circulating Biochemical Data

Blood samples were collected between 8 and 10 a.m. after an overnight fast and analyzed by the central laboratory of Kaohsiung Chang Gung Memorial Hospital to determine the serum levels of hemoglobin, total cholesterol, triglycerides, glycated hemoglobin (HbA1c), homocysteine, vitamin B12, and folate. The blood samples were centrifuged at 3000 rpm for 10 min to obtain serum samples, and all circulating biochemical markers were immediately analyzed, as reported previously [17,18,19].

### 2.4. Anthropometric Measurements

Height and weight were measured, with patients wearing a disposable gown given to them. The body mass index (BMI) was calculated as the weight in kilograms divided by the height in meters squared. The waist circumference (narrowest diameter between the xiphoid process and iliac crest) was measured using a Lufkin measuring tape. The hip circumference was measured at the level of maximum extension of the hip. 

### 2.5. Dietary, Exercise, and Constipation Assessments

With regard to the Mini-Nutritional Assessment (MNA) (maximum score: 30) [20], a score below 23.5 was defined as at risk and 24–30 was defined as a normal nutritional status. The patients and their caregivers also received specific dietary counseling on required dietary adaptations, proposals of menus, and advice. In this study, we evaluated their diets by making them complete a 3-day diary diet record (fluids included milk, juice, and soup). A single and well-trained dietitian provided oral instructions on how to perform the dietary registration and analysis. He presented them with a sample spoon, cup, plate, and bowel to permit them to report similar sizes and amounts. The dietary intake was recorded for two weekdays (from Monday to Friday) and one weekend day (Saturday or Sunday). Thereafter, patients and their caregivers were interviewed by the same dietitian to determine and correct the amount of ingested food using a food replica model (Nasco food replicas). 

We analyzed the macronutrients according to the information from the Taiwan Food and Drug Administration website [21]. The daily intake of energy, macronutrients, and micronutrients was calculated as the average of the 3-day food records. After this dietary assessment, patients and their caregivers received personalized dietary counseling and recommendations to improve their identified dietary inconsistencies. However, the changes in a patient’s dietary behavior were not assessed thereafter.

We modified a previous question that was used for a survey on exercise habits [22]. The patients and their caregivers were asked the following question during the interview: “Did you practice sports or physical exercise sufficient to produce sweating or shortness of breath?” Those who answered “≥2 h per week” were defined as having exercise habits.

The following mandatory information on constipation symptoms was collected by the case manager: loose stools rarely present without the use of laxatives; less than three bowel movements per week; manual maneuvers necessary to facilitate defecation; hard or lumpy stools; straining during defecation; the need to use laxatives and stool pattern; the sensation of incomplete evacuation; and the sensation of anorectal obstruction. Constipation is mainly diagnosed using the Rome III criteria, according to the patient and caregiver’s report [6,22,23]. 

### 2.6. Statistical Analysis

All values were expressed as the mean ± standard deviation. The chi-square test was used to compare categorical variables. Non-parametric methods were used when continuous variables were non-normally distributed. The Mann–Whitney U tests were used to compare the differences between two groups. We chose those factors with significant differences, including the MMSE, MNA, and water intake (*p* < 0.05), between two groups and the factors previously reported to be associated with constipation, including age, fiber, and exercise, for the binary logistic regression analysis. Multiple binary logistic regression was used to determine the factors associated with constipation in patients with dementia. Statistical analysis was performed using the SPSS version 22.0 software package (IBM, Armonk, NY, USA). A *p* value of <0.05 was considered significant.

## 3. Results

### 3.1. Clinical Characteristics and Demographic Data of All Participants 

The age, sex, MMSE, CDR, body height and weight, BMI, waist circumference, hip circumference, exercise habits, and comorbidities of the 119 patients with dementia are listed in Table 1. Of them, 30 (25.2%) experienced constipation. The hemoglobin, total cholesterol, triglyceride, HbA1c, homocysteine, vitamin B12, and folate levels are presented in Table 2. 

### 3.2. Nutrition Status and Diet Analysis in All Participants

The daily intake analysis conducted through diet diaries was recorded as the total caloric, carbohydrate, lipid, protein, calcium, fiber, salt (sodium), and water intake amounts (Table 2). The MNA was 23.80 ± 3.45, and 52 (43.7%) patients were at risk of malnutrition.

### 3.3. Clinical Characteristics and Demographic Differences in Patients with Constipation

We divided the patients into two groups, according to the status of constipation, and compared their basic demographic characteristics. No significant differences were found for age, gender, CDR, BMI, or the number of exercise habits between the two groups. However, MMSE was statistically higher in patients without constipation (20.4 ± 5.9) than in those with constipation (16.9 ± 7.7) (*p* = 0.028). The medication, which included acetylcholinesterase inhibitors (*p* = 0.152) and antipsychotic medications (*p* = 0.788), displayed no significant differences in the two groups. There were no significant changes in the clinical and demographic data, including diabetes mellitus, hypertension, coronary artery disease, hyperlipidemia, and chronic kidney disease, and biochemical data such as hemoglobin, total cholesterol, triglyceride, HbA1c, homocysteine, vitamin B12, and folate levels. MNA was statistically higher in patients without constipation (24.2 ± 3.5) than in those with constipation (22.6 ± 2.9) (*p* = 0.029). The number of patients at risk of malnutrition was higher in the no constipation group (60%) than in the constipation group (38.2%) (*p* = 0.029). In the daily intake analysis, the total caloric, carbohydrate, lipid, protein, calcium, fiber, and salt (sodium) levels were not significantly different in patients with constipation and in those without constipation (Table 3). However, the water intake was significantly lower in patients with constipation (546.81 ± 240.01) than in those without constipation (808.04 ± 404.24) (*p* = 0.001).

### 3.4. Risk Factors for Constipation in Patients with Dementia

The MMSE, MNA, and water intake were significantly lower in the constipation group (*p* < 0.05). Therefore, we chose the MMSE, MNA and water intake, and common risk factors associated with constipation, including age, fiber, and exercise, for the analysis (Table 4). Multiple binary logistic regression was performed to determine the risk factors for constipation in patients with dementia. We found that the amount of water was the only associated risk factor for constipation in patients with dementia (*p* = 0.002). However, fiber was not associated with constipation (*p* = 0.051).

## 4. Discussion

From a Taiwan dementia center survey that involved private in-person interviews with dietitian consultation, the prevalence of constipation symptoms (defined based on the stool consistency) (25.2%) in patients with dementia was higher than its overall prevalence. Constipation symptoms were defined based on stool consistency and frequency. Dementia patients with a lower MMSE score, risk of malnutrition, and lower water intake had a higher tendency to develop constipation. After controlling for other known factors for chronic constipation in adults, the only modifiable factor that may improve constipation was increasing the dietary intake of liquids, instead of fiber intake or exercise. 

Chronic constipation is a common geriatric syndrome. The prevalence rates of chronic constipation are as follows: In patients older than 60 years, 15–20% [24]; in patients older than 84 years, 20.0–37.3% [22,25]; and in patients receiving long-term care, 80% [22,25]. The mean age of our study group was 75.2 years, with a higher prevalence of constipation (25.2%) than that in normal older adults [24]. A similar study investigating constipation reported a prevalence of 24.5% in patients with Parkinson’s disease [26]. This means that chronic constipation is a frequent problem and needs to be addressed when caring for patients with dementia. 

In this study, the constipation group had lower MMSE scores (*p* < 0.05). This finding indicates that severe dementia may be associated with the risk of constipation. The correlation analysis revealed a positive correlation between dementia prevalence and constipation prevalence [27]. Based on the similarity of the prevalence distributions of these two diseases, it can be speculated that Alzheimer’s disease may be correlated with constipation, which means that either disease may share a common factor, such as a low fiber intake or decreased exercise, or one disease could contribute to the occurrence of the other disease [27].

Our study showed that patients with constipation (77.2 ± 8.3) were older than those without constipation (74.5 ± 7.4) (*p* = 0.090). In older adults, constipation can be the result of multiple factors, including common diseases that can cause chronic constipation, functional or organic diseases of the colorectum and anus, and drugs [28]. In patients with Parkinson’s disease, an older age and younger age of onset were associated with the incidence of constipation [26]. Although age is also an important factor, it could not be modified during the care of these patients. 

In addition to aging, the common risk factors associated with constipation included the following: (1) the fluid intake (daily total fluid intake >1.5 L); (2) the dietary fiber intake (>25 g/d); (3) activity (old patients, who are wheelchair-bound, who are immobile, or with impaired physical mobility); a lack of exercise for a long time; and slow intestinal peristalsis [28]. We also analyzed these three factors in our study to determine the most important factor in our patients. The third most important factor was activity. We used exercise to evaluate our patients. In our study, patients’ dementia was not severe. They had very mild dementia or mild dementia; as such, they could still express themselves well and exercise. However, there were no significant changes in exercise habits.

The MNA has recently been designed and validated to provide a single, rapid assessment of the nutritional status in older adults and to identify people at risk of malnutrition [29]. MNA was statistically higher in patients without constipation (24.2 ± 3.5) than in patients with constipation (22.6 ± 2.9) (*p* = 0.029). The number of patients with a risk of malnutrition was higher in patients without constipation (60%) than in those with constipation (38.2%) (*p* = 0.029). MNA is a questionnaire used for whole-body health evaluation, including body weight change, appetite, and body figure. In our study, we found that patients at risk of malnutrition according to their MNA scores were also at risk of constipation, and this finding was similar to what was reported for those with Parkinson’ s disease [30].

An increased intake of dietary fiber is widely considered to protect against constipation, and several studies have found an inverse relationship between dietary fiber intake and constipation. Fiber is important for stool passage. However, almost all older individuals in this study were unable to achieve the standard fiber intake. Studies have indicated that a high-fiber diet can increase the stool weight, resulting in a decreased colon transit time, while a poor-fiber diet induces constipation [31]. However, a high-fiber diet could improve symptoms in patients with normal colonic transit and anorectal function, while constipated patients with delayed colonic transit did not exhibit improvement by increasing their dietary fiber [32]. We found no significant differences between patients with dementia with and without constipation in terms of fiber intake in this study. Only one patient with constipation (3.3%) consumed >25 gm per day, while seven (7.9%) without constipation consumed >25 g per day. The fiber intake was generally low in our study. However, similar to the reports of other studies [33,34,35], we were unable to find such an association. This could be due to the fact that the dietary fiber intake of most study participants was too low to have a protective effect.

In our study, the water intake was significantly lower in patients with constipation (584.11 ± 277.60) than in those without constipation (788.60 ± 422.42) (*p* = 0.015). Furthermore, after adjusting for all possible factors, a low water intake was an independent risk factor associated with constipation in patients with dementia (*p* < 0.05). We further analyzed 14 patients with constipation (46.7%) who drank >500 mL and 66 (74.2%) without constipation who drank >500 mL per day (*p* = 0.006). A few studies have reported an association between the fluid intake and intestinal constipation. Epidemiological evidence has indicated an association between a lower fluid intake and intestinal constipation [36]. An inadequate fluid intake or excessive fluid loss due to diarrhea, vomiting, or febrile illness may cause hardening of the stool, and is considered to be an important cause of constipation, especially in infants [37]. Increasing one’s liquid intake is commonly recommended for constipated children, adults, and older adults [10,11,12,22,38,39,40]. A lower frequency of constipation among individuals with a higher fluid intake was observed (20%, 15%, or 11%, respectively) [38]. The stool frequency and stool weight were significantly decreased during the liquid deprivation week, despite the absence of changes in the mean oral–anal transit time. The effects of the fluid intake on constipation have not been fully elucidated [10,11,12,40,41]. In older adults, a low liquid intake may result in hypohydration, which often causes constipation [41]. Poor dentition, a decreased production of saliva, and dehydration, especially in the warmer months, make the problem more complex [40,41]. Further clinical trials and epidemiological studies that consider the international recommendations for fluid intake for older adults are required. The importance of an increased fluid intake was questioned by the European Society for Pediatric Gastroenterology, Hepatology, and Nutrition/North American Society for Pediatric Gastroenterology, Hepatology, and Nutrition. A clinical trial in children reported that only an increased fluid intake with or without behavioral intervention had no significant effect [42,43]. No randomized controlled trials have evaluated the benefit of water supplementation alone in patients with constipation. However, water supplementation of 1.5–2 L per day improved the stool frequency [43] and decreased the need for laxatives [44,45,46]. Therefore, a higher water intake may be beneficial for the prevention and treatment of mild intestinal constipation. Further studies are required to better understand the role of water and fluids in the etiology and treatment of intestinal constipation in patients with dementia.

The major strength of this investigation is that it is the first study to investigate the risk factor of constipation in patients with dementia who were diagnosed by two neurologists or a psychologist’s confirmation. Some risk factors were found. However, there were some limitations in this study. The first limitation of our study was the cross-sectional design of the study; we cannot infer the cause and outcome relationship. The second limitation of the study is that the incidence could be underestimated because of dementia and information from the caregiver. Third, this study did not consider the amount of water present in food, but only in the form of liquids. The fourth limitation is the dietary data collection having possible recall bias, despite the double confirmation by caregivers. The fifth limitation is that we did not measure serum osmolality for dehydration. The final limitation is the small number of participants. This indicates that a large sample size is needed to show the effect of fluid intake on constipation. In future studies, prospective randomized clinical trials with water ingestion to prevent dehydration should be performed to confirm the results.

## 5. Conclusions

A low fluid intake was associated with constipation. An increased adequate water intake could improve constipation. However, randomized prospective clinical trials are still needed to confirm these results. 

## Figures and Tables

**Table 1 ijerph-17-09006-t001:** Demographic and clinical characteristics of patients with dementia.

	All Participants (*n* = 119)
Age, years	75.2 ± 7.7
Gender (men/women)	53/66
Mini-Mental State Examination	19.5 ± 6.6
Clinical dementia rating (0.5/1)	82/37
Body height, cm	154.3 ± 8.3
Body weight, kg	58.7 ± 10.5
Body mass index (kg/m^2^)	24.6 ± 3.9
Waist circumference, cm	86.2 ± 11.5
Hip circumference, cm	94.9 ± 9.7
Waist/hip ratio	0.091 ± 0.07
Exercise habit, *n* (%)	77 (64.7%)
**Comorbidity**	
Diabetes mellitus, *n* (%)	38 (31.9%)
Hypertension, *n* (%)	62 (52.1%)
Hyperlipidemia, *n* (%)	24 (20.1%)
Coronary artery disease, *n* (%)	19 (15.9%)
Chronic kidney disease, *n* (%)	8 (6.7%)
**Constipation**	
Patients with constipation, *n* (%)	30 (25.2%)

Note: Mean ± standard deviation.

**Table 2 ijerph-17-09006-t002:** Biochemical characteristics and daily intake of patients with dementia.

	All Participants (*n* = 119)
**Biochemistry**	
Hemoglobin, g/dL	13.07 ± 1.83
Total cholesterol, mg/dL	184.79 ± 41.32
Triglycerides, mg/dL	119.66 ± 69.17
HbA1c, %	6.19 ± 1.62
Homocysteine, umole/L	17.03 ± 15.99
Vitamin B12, pg/mL	776.09 ± 792.91
Folate, ng/mL	11.81 ± 9.39
Mini-Nutritional Assessment	23.80 ± 3.45
Number of with risk malnutrition, *n* (%)	52 (43.7%)
Daily intake	
Total caloric, kcal	1507.34 ± 586.40
Carbohydrate, as % of total energy	60.24 ± 11.98
Lipid, as % of total energy	26.38± 10.29
Protein, as % of total energy	14.21 ± 11.98
Calcium, mg	390.25 ± 265.48
Fiber, gm	13.22 ± 8.81
Salt (sodium), mg	853.29 ± 755.67
Water, mL	742.19 ± 386.01

Note: Mean ± standard deviation. Mini-Nutritional Assessment scores below 23.5 were defined as being at risk and 24–30 was defined as a normal nutritional status.

**Table 3 ijerph-17-09006-t003:** Clinical and biochemical characteristics and the daily intake subgroup analysis.

	Patients with Constipation (*n* = 30)	Patients without Constipation (*n* = 89)	*p*-Value
Age, years	77.2 ± 8.3	74.5 ± 7.4	0.090
Gender (M/F)	16/14	37/52	0.134
Mini-Mental State Examination	16.9 ± 7.7	20.4 ± 5.9	0.028 *
Clinical dementia rating (0.5/1)	18/12	64/25	0.082
Body mass index (kg/m^2^)	24.8 ± 3.8	24.6 ± 4.0	0.877
Exercise habit	17 (56.7%)	60 (67.4%)	0.178
Medication			
Acetylcholinesterase inhibitors	14 (46.7%)	33 (37.1%)	0.152
Antipsychotic medications	2 (6.7%)	7 (7.9%)	0.788
**Comorbidity**			
Diabetes mellitus, %	12 (40%)	26(29.2%)	0.161
Hypertension, %	18 (60%)	44 (49.4%)	0.295
Hyperlipidemia, %	5 (16.7%)	19 (21.3%)	0.427
Coronary artery disease, %	6 (20%)	13 (14.6%)	0.306
Chronic kidney disease, %	2 (6.7%)	6 (6.7%)	0.630
**Biochemistry**			
Hemoglobin, g/dL	12.60 ± 1.91	13.23 ± 1.79	0.106
Total Cholesterol, mg/dL	174.55 ± 40.30	188.21 ± 41.33	0.124
Triglycerides, mg/dL	121.37 ± 81.87	119.08 ± 64.85	0.876
HbA1c, %	6.21 ± 1.20	6.18 ± 1.75	0.927
Homocysteine, umole/L	19.97 ± 9.94	15.99 ± 17.58	0.251
Vitamin B12, pg/mL	619.07 ± 457.01	830.88 ± 876.13	0.209
Folate, ng/mL	11.82 ± 10.46	11.82 ± 9.06	0.999
Mini-Nutritional Assessment	22.6 ± 2.9	24.2 ± 3.5	0.029 *
Number of with risk malnutrition, *n* (%)	18 (60%)	34 (38.2%)	0.031 *
**Daily intake**			
Total caloric, kcal	1552.80 ± 636.82	1492.02 ± 571.41	0.625
Carbohydrate, as % of total energy	61.11 ± 13.70	59.94 ± 11.41	0.678
Lipid, as % of total energy	14.23 ± 4.48	14.20 ± 4.14	0.974
Protein, as % of total energy	25.18 ± 11.53	26.69 ± 9.88	0.462
Calcium, mg	353.34 ± 249.71	402.84 ± 270.87	0.380
Fiber, gm	11.90 ± 5.99	13.67 ± 9.56	0.343
Salt (Sodium), mg	955.28 ± 1115.80	818.91 ± 592.19	0.527
Water, mL	546.81 ± 240.01	808.04 ± 404.24	0.001 *

Note: Mini-Nutritional Assessment scores below 23.5 were defined as being at risk and 24–30 was defined as a normal nutritional status. A chi-square test was used to compare categorical variables. Mann–Whitney U tests were applied to compare the differences between groups. Significance was set at *p* < 0.05. * *p* < 0.05.

**Table 4 ijerph-17-09006-t004:** Binary logistic regression models for constipation in patients with dementia.

	Odds Ratio	95% Confidence Interval	*p*-Value
Age	0.998	0.932–1.067	0.944
Mini-Mental State Examination (MMSE)	1.048	0.968–1.135	0.250
Water, mL	1.004	1.001–1.007	0.002 *
Mini-Nutritional Assessment Score (MNA)	1.067	0.917–1.241	0.404
Fiber	1.009	0.953–1.001	0.051
Exercise (category)	0.935	0.331–2.642	0.899
Constant	0.068		0.438

Note: Multiple binary logistic regression was used to determine the factors, including age, exercise, MMSE, MNA, and water intake, associated with constipation in patients with dementia. Multiple binary logistic regression was performed by the enter method. * *p* < 0.05.

## Data Availability

The datasets generated during and/or analyzed during the current study are available from the corresponding author (Nai-Ching Chen) on reasonable request.

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
