# Peer review of "Constipation and Its Associated Factors among Patients with Dementia"

_ijerph, 2020, doi:10.3390/ijerph17239006_

Round 1

Reviewer 1 Report

Chen et al. present a cross-sectional study in which they describe the praevalence of constipation in patients with dementia. Although they make a case for the importance of their results, there are several issues in the report that need to be addressed.

1. the frequency of constipation. While the authors state that 25% of patients are constipated in table 1, they write in their Limitation section that all patients fullfilled the ROMEIII criteria for constipation. According to the method section "Constipation is mainly diagnosed using the Rome III criteria according to the patient’s self-Report " Now the question is, how did the authors arrive at 25% constipated patients?

2. Is a food recall study appropriate in patients with dementia? Did the patients have help from a relative to complete the study related questionnaires? According to the table 3,  MMSE was around 20 Points. How confident are the authors that the obtained data is reliable? More over, since patients with constipation had a lower MMSE score, might this result be biased?

3. Information about the patients medication is missing. Without this information, a conclusion about the risk factors for constipation should not be reached.

4. The Authors talk about a comprehensive battery of tests to assess the congnitive ability of all patients. What did the authors refer to with this statement?

Author Response

  1. the frequency of constipation. While the authors state that 25% of patients are constipated in table 1, they write in their Limitation section that all patients fullfilled the ROMEIII criteria for constipation. According to the method section "Constipation is mainly diagnosed using the Rome III criteria according to the patient’s self-Report " Now the question is, how did the authors arrive at 25% constipated patients?

Response:

The history of constipation was confirmed by the caregivers.

The included criteria in the study were that the patients with their caregivers could receive all examinations and complete the records. The main caregiver needs to stay with the patient every day and have no subjective cognitive decline/complain. (line92-94)

However, we added the dementia of the patients as a limitation.

The second limitation of the study is that 25% of the participants fulfilled the Rome III criteria despite confirmations by caregivers. However, the incidence could be underestimated because of dementia. (line294-296)

  1. Is a food recall study appropriate in patients with dementia? Did the patients have help from a relative to complete the study related questionnaires? According to the table 3, MMSE was around 20 Points. How confident are the authors that the obtained data is reliable? Moreover, since patients with constipation had a lower MMSE score, might this result be biased?

Response:

The included criteria in the study were the patients with their caregivers could receive all examinations and complete the records. The main caregiver needs to stay with the patient every day and have no subjective cognitive decline/complain. (line92-94)

In our country, patients suffered from any cognitive problem was not allowed to go to hospital alone, because their family will worry about their safety even the disease is very mild.

We added the dementia related effects as a limitation.

The fourth limitation is the dietary data collection with possible recall bias despite the double confirmation by caregivers. (line298-299)

  1. Information about the patient’s medication is missing. Without this information, a conclusion about the risk factors for constipation should not be reached.

Response:

We add the medications include acetylcholinesterase inhibitors and antipsychotic medications into revised manuscript for possible drugs effect in patient with dementia. The medication included acetylcholinesterase inhibitors (p=0.152) and antipsychotic medications (p=0.788) had no significant difference in two groups. (line 177-178)

  1. The Authors talk about a comprehensive battery of tests to assess the cognitive ability of all patients. What did the authors refer to with this statement?

Response:

In our country, the diagnosis of the dementia needs to complete the examination included neurologic history and examination, a brain image, the Mini-Mental State Examination (MMSE), Clinical Dementia Rating (CDR) Scale, blood examination. Furthermore, the diagnosis needs to be confirmed by another neurologist or psychologist.

A trained neuropsychologist administered the neurobehavioral tests, including the Mini-Mental State Examination (MMSE) [15] to assess their general intellectual function and clinical dementia rating (CDR) scores [16] to evaluate the severity of patients’ cognitive impairment in the dementia integrated outpatient clinic. (line103-106)

Reviewer 2 Report

I would like to thank the authors for this interesting article. Nevertheless, the manuscript needs some enhancements and mainly at the level of the statistical analyses and the discussion.

Please find below my comments regarding each section.

Introduction:

  • Introduction should be enhanced. The knowledge gap is not very convincing and strong. 
  • The authors focus on constipation towards the end of the introduction while constipation is their main variable. I believe it deserves more attention in the intro. 
  • Risk factors of constipation should be more emphasized especially in the context of dementia.

Methods:

  • Why were the participants at the outpatient clinic? what are the reasons of their admission?
  • Inclusion/exclusion criteria based on the cognitive level are not clear. The authors should also specify when a caregiver was responsible for data collection and when the actual patient filled the 24- hrs dietary recalls (and what was their cognitive level)
  • L112: there is a mistake in the description of data collection. it should be 2 weekdays and 1 weekend day
  • L123: how was a small sample size defined?
  • More details about the statistical analyses are needed. How did you perform the analysis? how did you choose which variables are entered in the models? you did binary logistic regressions with each variable? if yes, did you adjust your p for multiple testing?

Results:

  • In table 2, it would be more informative if you report the results of macronutrients as % of total energy. Same comment for table 3.
  • Based on what you decided to enter the variables in the logistic models? (table 4).  

Discussion:

  • L179-186: It is not clear if these results are for the current study. This paragraph needs some enhancements. 
  • L255: why did you reference pediatric associations? knowing that your sample consists of older adults?
  • L264: what about limitations pertaining to dietary data collection? 
  • L271: what are the strengths of this study?
  • In general, you did not tackle the mechanisms describing the association between dementia and constipation.
  • What are the implications of this study? future perspectives?

Author Response

Introduction:

Introduction should be enhanced. The knowledge gap is not very convincing and strong. The authors focus on constipation towards the end of the introduction while constipation is their main variable. I believe it deserves more attention in the intro. Risk factors of constipation should be more emphasized especially in the context of dementia.

Response:

We revised our introduction and emphasized in the important and risk factors of constipation in patients with dementia.

Caring for the patients with dementia poses many challenges for families and caregivers. A meta-analysis study found that dementia family caregivers were significantly more stressed than non-dementia caregivers and more likely to experience serious depressive symptoms and physical problems [2]. Overall, one-third of the caregivers (disregarding care demand and care recipient in dementia stage) provided >14 hours of care on a daily basis. Meanwhile, basic and instrumental activities of daily living (ADLs) impairments (e.g., bathing and toileting in daily basic life need) (68%) were the most common concerns of most caregivers [3]. Understand how to care the daily activities of the dementia patients could be helpful to families, caregivers and patients themselves.

Toileting, which includes bowel movement (stool passage) and urination are basic needs, and constipation and urinary incontinence are common health problems among older adults including those with dementia. Constipation is one of the most common health problems encountered in primary healthcare settings and occurring in the general population as well as among older adults, especially those with dementia [3-5]. Constipation is characterized by formation of hard stools, infrequent defecation (fewer than two times per week), and prolonged and difficult evacuation of stools. Patients with persistent constipation have a lower quality of life and experience financial burden owing to health costs [6-9]. Patients with dementia might not be able to express the pain and discomfort of the constipation orally and so can be assumed that aggression in patients with dementia and so the constipation will be untreated or wrongly given an anti-psychotic drug for these symptoms [10-12]. This can in turn make matters worse because the anti-psychotic drugs could cause constipation. Hence, how to prevent constipation is very important. An ounce of prevention is worth a pound of cure. (line46-67)

Methods:

Why were the participants at the outpatient clinic? what are the reasons of their admission?

Response:

We want to investigate patients with dementia in stable condition. We choose outpatient rather than admission participants.

Inclusion/exclusion criteria based on the cognitive level are not clear. The authors should also specify when a caregiver was responsible for data collection and when the actual patient filled the 24- hrs dietary recalls (and what was their cognitive level)

Response:

The included criteria in the study were that the patient with their caregivers could receive all examinations, records and complete the records. The main caregiver needs to stay with the patient every day and have no subjective cognitive decline/complain. Therefore, we add the limitation included “The second limitation of the study is that 25% of the participants fulfilled the Rome III criteria despite confirmations by caregivers. However, the incidence could be underestimated because of dementia and information from caregiver.” (line294-297)

“The fourth limitation is the dietary data collection with possible recall bias despite the double confirmation by caregivers.” (line298-299)

L112: there is a mistake in the description of data collection. it should be 2 weekdays and 1 weekend day

Response:

Dietary intake was recorded for 2 weekdays (from Monday to Friday) and one weekend day (Saturday or Sunday). line126-127)

L123: how was a small sample size defined?

More details about the statistical analyses are needed. How did you perform the analysis? how did you choose which variables are entered in the models? you did binary logistic regressions with each variable? if yes, did you adjust your p for multiple testing?

Response:

According to G-power software, we choose the statistic method of Linear multiple regression in F test: Fixed model, R2 deviation from zero. We fill in α=0.05, power=0.8, number of predictors = 5. Then we fill in squared multiple correlation ρ2=0.16, then we use calculate. We get the number of Effect size f2. Finally, we use this number to calculate and transfer to main window. We use the calculate again and we get the less sample size as 74. (the minimum number)

Non-parametric methods were used when continuous variables were non-normally distributed. The Mann-Whitney U tests were used to compare the differences between two groups. (line148-149)

In our study, we collect 119 patients with dementia with several variable but only 6 possible predictors. Our sample size is bigger than 74.

We choose those factors with significant difference included MMSE, MNA, water intake (p<0.05) between two groups and the factors previous reported to be associated with constipation included age, fiber and exercise into binary logistic regression analysis. Multiple binary logistic regression was used to determine the factors associated with constipation in patients with dementia. (line150-154)

Results:

In table 2, it would be more informative if you report the results of macronutrients as % of total energy. Same comment for table 3.

Response:

We revised the results of macronutrients as % of total energy in revised table 2 and 3.

Based on what you decided to enter the variables in the logistic models? (table 4). 

Response:

We choose those factors with significant difference included MMSE, MNA, water intake (p<0.05) between two groups and the factors previous reported to be associated with constipation included age, fiber and exercise into binary logistic regression analysis. Multiple binary logistic regression was used to determine the factors associated with constipation in patients with dementia. (line150-154)

Discussion:

L179-186: It is not clear if these results are for the current study. This paragraph needs some enhancements.

Response:

The major strength is that the first study to investigate the risk factor of constipation in the patients with dementia which were confirmed by two neurologists or psychologist’s confirmation. Some risk factors were found. (line291-293)

L255: why did you reference pediatric associations? knowing that your sample consists of older adults?

Response:

We add the reference which consisted of older adults in revised manuscript. (references 10-12)

L264: what about limitations pertaining to dietary data collection?

Response:

However, there are some limitations in this study. The first limitation of our study was the cross-sectional design of the study; we cannot infer the cause and outcome relationship. The second limitation of the study is that 25% of the participants fulfilled the Rome III criteria despite confirmations by caregivers. However, the incidence could be underestimated because of dementia and information from caregiver. Third, this study did not consider the amount of water present in food, but only in the form of liquids. The fourth limitation is the dietary data collection with possible recall bias despite the double confirmation by caregivers. The fifth limitation is that we did not measure serum osmolality for dehydration. The final limitation is the small number of participants. This indicates that a large sample size is needed to show the effect of fluid intake on constipation. In the future study, prospective randomized clinical trials with water ingestion to prevent dehydration should be performed to confirm the results. (line293-303)

L271: what are the strengths of this study?

In general, you did not tackle the mechanisms describing the association between dementia and constipation.

What are the implications of this study? future perspectives?

Response:

The pathogenesis of constipation is multifactorial, with a focus on dietary behavior such as low fiber consumption, lack of adequate fluid intake, lack of mobility, disturbance in hormone balance, side effects of medications with anti-cholinergic actions, or hemorrhoids [8-12]. However, constipation has never been isolated in patients with dementia. Given that epidemiological and clinical studies reported the association of constipation in patients with dementia and that little is known about the risk factors of constipation such as demographic factors and diet behaviors. (line 68-73)

The major strength is that the first study to investigate the risk factor of patients with dementia which were confirmed by two neurologists or psychologist’s confirmation. Some risk factors were found. (line291-293) In the future study, prospective randomized clinical trials with water ingestion to prevent dehydration should be performed to confirm the results. (line301-302)

Reviewer 3 Report

Thank you for this interesting article.

Abstract:

  1. Identify the sample size and over what time frame the data was collected
  2. How was dementia severity determined?
  3. Clarify that patients are out patients

Introduction

  1. Line 62 – why is disturbance in hormone balance italicized?

Methods

  1. Provide further detail on how the participants were instructed to complete the diet record. For example were standardized forms used?
  2. How was water intake determined? Fluids only or also including food? Was a nutrient analysis program used?
  3. Comment on the multivariate analysis strategy considering the low number of cases. It is unclear what variables were entered into the model. Why was MNA absolute score and the malnutrition cut point both used? These would be collinear. Revisit the multivariate analyses.

Results

  1. Fibre intake is generally low and there was a nonsignificant lower intake in those with constipation. Should the only recommendation be to increase fluid intake? Fibre needs to be increased too based on these results. As noted in the Discussion, lack of statistical difference is likely due to low intake.

Discussion

  1. Lines 179-80 – how was incidence vs. prevalence of constipation determined?
  2. Line 184- based on table 4, not all potential causes of constipation were controlled for. For example, fibre and exdercise were not included in the model in Table 4.

Limitations need to be further elaborated

  • Biochemistry was measured. The authors could have measured serum osmolality for dehydration.
  • Due to the small sample size, only a few variables could be considered in the multivariate analyses; a fully adjusted model was not feasible.
  • Sample is not generalizable.

Author Response

Abstract:

Identify the sample size and over what time frame the data was collected

Response:

We revised the abstract and add sample size and time frame in the new manuscript.

This cross-sectional study was conducted at the Chang Gung Memorial Hospital-Kaohsiung since 2019-January to 2019-November, a medical center and the main referral hospital that serves 3 million inhabitants of southern Taiwan. 119 patients with dementia were evaluated using the Rome III diagnostic criteria for functional constipation. There were 30 patients with dementia divided into constipation group and 89 patients with dementia divided into no constipation group. (line25-30)

How was dementia severity determined?

Response:

Clinical dementia rating scores is used to evaluate the severity in patients with dementia of the outpatient clinic. (line 31-32)

Clarify that patients are out patients

Response:

Clinical dementia rating scores is used to evaluate the severity in patients with dementia of the outpatient clinic. (line 31-32)

Introduction

Line 62 – why is disturbance in hormone balance italicized?

Response:

Thank you for you reminds.

Methods

Provide further detail on how the participants were instructed to complete the diet record. For example, were standardized forms used?

Response:

With regard to Mini-Nutritional Assessment (MNA) (maximum score: 30) [20], a score below 23.5 was defined as at risk, and 24–30 as normal nutritional status. The patients and their caregivers also received specific dietary counseling on required dietary adaptations, proposals of menus, and advice. In this study, we evaluated their diets by making them complete a 3-day diary diet record (included milk, juice, and soup…). A single and well-trained dietitian provided oral instructions on how to perform the dietary registration and analysis. He gave them the sample of spoon, cup, plate and bowel to let them have similar size and amount report. Dietary intake was recorded for 2 weekdays (from Monday to Friday) and one weekend day (Saturday or Sunday). Thereafter, patients and their caregivers were interviewed by the same dietitian to determine and correct the amount of ingested food using a food replica model (Nasco food replicas). We analysis the macronutrients according to the information from Taiwan Food and Drug Administration website (https://consumer.fda.gov.tw/Food/TFND.aspx?nodeID=178). (line120-131)

How was water intake determined? Fluids only or also including food? Was a nutrient analysis program used?

Response:

With regard to Mini-Nutritional Assessment (MNA) (maximum score: 30) [20], a score below 23.5 was defined as at risk, and 24–30 as normal nutritional status. The patients and their caregivers also received specific dietary counseling on required dietary adaptations, proposals of menus, and advice. In this study, we evaluated their diets by making them complete a 3-day diary diet record (fluids included milk, juice, and soup…). (line120-124)

Comment on the multivariate analysis strategy considering the low number of cases. It is unclear what variables were entered into the model. Why was MNA absolute score and the malnutrition cut point both used? These would be collinear. Revisit the multivariate analyses.

Response:

Thanks for you remind. We delete the malnutrition cut point.

Results

Fibre intake is generally low and there was a nonsignificant lower intake in those with constipation. Should the only recommendation be to increase fluid intake? Fibre needs to be increased too based on these results. As noted in the Discussion, lack of statistical difference is likely due to low intake.

Response:

After we add fiber and exercise into regression model in Table 4.

We choose those factors with significant difference included MMSE, MNA, water intake (p<0.05) between two groups and the factors previous reported to be associated with constipation included age, fiber and exercise into binary logistic regression analysis. Multiple binary logistic regression was used to determine the factors associated with constipation in patients with dementia. (line150-154)

We found no significant differences between patients with dementia with and without constipation about fiber intake in the study. Only one patient with constipation (3.3%) consumed >25 gm per day, while seven (7.9%) without constipation consumed >25 g per day. The fiber intake is generally low in our study. However, similar to the reports of other studies [33-35], we were unable to find such an association. This could be due to the fact that the dietary fiber intake of most study participants was too low to have a protective effect.

(line 257-262)

Discussion

Lines 179-80 – how was incidence vs. prevalence of constipation determined?

Response:

We corrected it as prevalence of constipation.

Line 184- based on table 4, not all potential causes of constipation were controlled for. For example, fibre and exercise were not included in the model in Table 4.

Response:

MMSE, MNA and water intake was significant lower in constipation group (p < 0.05). Therefore, we put MMSE, MNA, water intake and common risk factor associated with constipation age, fiber and exercise into analysis. Multiple binary logistic regression by enter method was performed to determine the risk factors for constipation in patients with dementia. We found that the amount of water was the only associated risk factors for constipation in patients with dementia (p = 0.002). However, fiber was not associated with constipation (p = 0.051). (line194-199)

Limitations need to be further elaborated

Biochemistry was measured. The authors could have measured serum osmolality for dehydration.

Response:

We add the lack of unmeasured serum osmolality for dehydration in our limitation section.

The fifth limitation is that we did not measure serum osmolality for dehydration. (line299-300)

Due to the small sample size, only a few variables could be considered in the multivariate analyses; a fully adjusted model was not feasible. Sample is not generalizable.

Response:

According to G-power software, we choose the statistic method of Linear multiple regression in F test: Fixed model, R2 deviation from zero. We fill in α=0.05, power=0.8, number of predictors = 5. Then we fill in squared multiple correlation ρ2=0.16, then we use calculate. We get the number of Effect size f2. Finally, we use this number to calculate and transfer to main window. We use the calculate again and we get the less sample size as 74. (the minimum number)

Non-parametric methods were used when continuous variables were non-normally distributed. The Mann-Whitney U tests were used to compare the differences between two groups. (line148-149)

In our study, we collect 119 patients with dementia with several variable but only 6 possible predictors. Our sample size is bigger than 74.

In our study, we collect 119 patients with dementia with several variables.  

We choose those factors with significant difference included MMSE, MNA, water intake (p < 0.05) between two groups and the factors previous reported to be associated with constipation included age, fiber and exercise into binary logistic regression analysis. Multiple binary logistic regression was used to determine the factors associated with constipation in patients with dementia. (line150-154)

Round 2

Reviewer 3 Report

Line 25 Bowel should be 'bowl'. In the limitations section, incidence should be changed to 'prevalence' (line 294)

Other requested edits have been made and are statisfactory.

Author Response

Reviewer 3

Line 25 Bowel should be 'bowl'. In the limitations section, incidence should be changed to 'prevalence' (line 294)

Response:

We revised the wrong words Bowel should be 'bowl'. (line 125) In the limitations section, incidence was changed to 'prevalence' (line 294).
